# OpenReview forum: "System Prompt Poisoning: Persistent Attacks on Large Language Models Beyond User Injection"
_ICLR.cc/2026/Conference — ICLR 2026 Conference Withdrawn Submission_

### Official Review · Reviewer_nDRW · 2025-10-28

**Soundness:** 3
**Presentation:** 3
**Contribution:** 2
**Rating:** 2
**Confidence:** 3

**Summary:**

This paper proposed the novel poison attacks against system prompts named System Prompt Poisoning (SPP). Specically, the paper formalizes SPP as an attack scenario in which an adversary modifies the system prompt to insert malicious or disruptive instructions that persist across sessions. The authors propose three implementation strategies, including brute-force poisoning, adaptive in-context learning poisoning, and adaptive chain-of-thought poisoning. Based on that, the authors introduce an automated attack framework called Auto-SPP to operationalize these techniques. Experimental results demonstrate the effectiveness of the proposed attacks.

**Strengths:**

- Well-written paper
- Comprehensive evaluation
- Clear empirical findings

**Weaknesses:**

- Unrealistic threat model
- Lack of causal distinction from existing prompt-injection taxonomy

**Questions:**

The authors propose the the system prompt poisoning attacks in this paper. The adversary can poison the system prompt to make the language models output random or harmful resposnes. The paper is well-written and experimentally thorough. However, I do have serveral comments on the threat model and the motivation of the proposed attacks.

- The proposed System Prompt Poisoning (SPP) assumes that an attacker can directly access and modify the system prompt that governs the LLM’s core behavior. However, in realistic deployment scenarios, this component is stored server-side or within a controlled application layer, and cannot be altered by end users or external adversaries. This assumption effectively grants the attacker full administrative control over the system—under which any attack becomes trivial. As such, the proposed “new attack class” is better understood as a case of configuration tampering, not an emergent or externally exploitable LLM vulnerability.

- Moreover, the exiting prompt injection attacks are also try to inject the poisoned prompt into the context of the LLMs, whose principle is the same as this paper. However, poison the memory/rag/outside input is possible while poison the system prompt directly is much more harder in practice. Therefore, the authors need to explain the fundamental differences between their proposed new attack and the old attack

---

### Official Review · Reviewer_eDMH · 2025-10-30

**Soundness:** 2
**Presentation:** 2
**Contribution:** 1
**Rating:** 2
**Confidence:** 4

**Summary:**

The paper proposes an attack which is called system prompt poisoning, where an attacker injects malicious instructions into the system prompt to override the original goal of the application (for example, to produce wrong outputs). The contradiction between the original system prompt and the malicious part degrades the performance on tasks.

**Strengths:**

The paper experiments with multiple datasets and shows an interactive setup with investigating the number of turns and how persistent the attack is.

**Weaknesses:**

- The threat model considered in this paper is not convincing. Usually, system prompts are set by the developer. It is not very realistic to assume they can be manipulated.

- Additionally, the system prompts used in this work are quite short. They are not stealthy, and an observer (the system developer who can inspect the system) can clearly detect that they are malicious.

- The output of the model is also clearly wrong due to the attack. If the application is deployed to do a narrow task, such as sentiment analysis, as the experiments in the paper show (figure 3), then a quick inspection of the quality of the output (or a test on examples with ground truth) would reveal that the system is not working as intended. This further makes the attack trivial to detect.

- Setting aside how trivial the attack is to detect and also whether it is realistic or not, the user can use various forms of prompt injection attack style (such as ignore your previous instructions) to mitigate the effect of the malicious system prompt. Prompt injection is unsolved yet, so it might be used as a defense/mitigation against this threat of system prompt poisoning.

- The paper mentions stealthiness against attacks, but I believe stealthiness is not the right word to use here because it implies that the attack is hard to detect. The paper does not experiment with any detection methods and the attack is relatively easy to detect.

- Regarding technical contribution, the methods used to poison system prompts are largely used for prompt injection.

- Minor conceptual error: the paper categorizes the prompts into system and user only. In practice, there are other prompts such as tool output. It mentions that most previous work investigates attacks that are coming from the user prompt, but indirect prompt injection, which is a major attack vector, is coming from tools output.

- The paper motivates the attack that it is a low cost ("This makes it a highly practical, low-cost attack."). This does not make a lot of sense regarding the threat model because the model developer is running the system so any cost will be incurred to the developer not the attacker.

- I believe the conclusion of the paper to highlight contradictions between system prompts and user prompts as a defense to system prompt poisoning is highly unrealistic. It is largely a common practice to assume that the system prompt is more trusted. This makes sense because 1) it is controlled, 2) it is not widely accessible, 3) it is easy to inspect in comparison to infinite options of user prompts. Putting higher trust in the user prompt opens the door to many other attacks.

Overall, I believe the threat model and the attack might be realistic and plausible in practice if 1) the system prompt is excessively long such that it is hard to inspect, 2) the malicious parts are hidden and can't be directly observed by human inspection, or 3) the attack is not so easy to detect from the output (e.g., because it is doing a side malicious task, or similar to backdoor literature, the output is wrong on narrower/more targeted cases).

**Questions:**

- What is the explicit/implicit nature of system prompts mentioned here "To evaluate the impact of SPP, we consider four attack scenarios combining the specification nature (explicit/implicit) and means (API/interactive) of system prompts (as detailed in Appendix C)"?

- What is the reason behind having a benign system prompt, followed by "update", given that the entire text here would be given to the system prompt directly?

---

### Official Review · Reviewer_Wd8u · 2025-10-31

**Soundness:** 3
**Presentation:** 3
**Contribution:** 3
**Rating:** 6
**Confidence:** 3

**Summary:**

This paper introduces "System Prompt Poisoning" (SPP), a novel attack that compromises LLMs by injecting malicious content into system prompts. The authors propose three poisoning strategies (brute-force, adaptive in-context, adaptive chain-of-thought) and develop Auto-SPP, an automated framework for generating poisoned prompts. Through comprehensive evaluation across four LLMs and two domains (mathematics and coding), the paper demonstrates SPP's severe effectiveness, persistence through long conversations, robustness against user-side defenses, and ability to bypass existing black-box detection mechanisms.

**Strengths:**

1. It addresses a critical gap in LLM security by focusing on system prompt vulnerabilities—an understudied area compared to user prompt injection—and formally defines SPP to distinguish it from existing threats.

2. The experimental design is comprehensive, covering diverse models, scenarios, and domains, with rigorous metrics (task degradation rate, execution cost) that effectively demonstrate SPP’s practical risks.

3. The Auto-SPP framework enhances real-world relevance by automating attack implementation, supported by cost analyses to contextualize its feasibility for different use cases.

**Weaknesses:**

1. The literature review in the related work section lacks depth, with limited discussion of recent defenses for system prompt security (e.g., prompt isolation mechanisms in some commercial LLMs) and insufficient comparison with similar poisoning attacks (e.g., training data poisoning), which weakens the positioning of SPP within the broader LLM security landscape.

2. Proposed defenses (e.g., system prompt integrity monitoring) remain conceptual—no lightweight prototype (e.g., a fast checksum-based verification tool for API scenarios) is provided, nor is the potential performance overhead of these defenses quantified, making it hard to assess their practicality.

3. The evaluation is limited to tasks with objective correctness (mathematics, coding); it does not test SPP on subjective tasks (e.g., sentiment analysis, creative writing), and the criteria for defining "successful poisoning" in such subjective contexts are not explored.

**Questions:**

None

---

### Official Review · Reviewer_5j99 · 2025-11-01

[review text omitted: it was posted to a different submission]

---

### Note · Authors · 2025-11-12

I have read and agree with the venue's withdrawal policy on behalf of myself and my co-authors.